

# Assessing distinct patterns of cognitive aging using tissue-specific brain age prediction based on diffusion tensor imaging and brain morphometry

Geneviève Richard[1,2,3], Knut Kolskår[1,2,3], Anne-Marthe Sanders[1,2,3], Tobias Kaufmann[1], Anders Petersen[4], Nhat Trung Doan[1], Jennifer Monereo Sánchez[1], Dag Alnæs[1], Kristine M. Ulrichsen[1,2,3], Erlend S. Dørum[1,2,3], Ole A. Andreassen[1], Jan Egil Nordvik[3] and Lars T. Westlye[1,2]

[1] NORMENT, KG Jebsen Centre for Psychosis Research, Division of Mental Health and Addiction, Oslo University Hospital & Institute of Clinical Medicine, University of Oslo, Oslo, Norway
[2] Department of Psychology, University of Oslo, Oslo, Norway
[3] Sunnaas Rehabilitation Hospital HT, Nesodden, Norway
[4] Center for Visual Cognition, Department of Psychology, University of Copenhagen, Copenhagen, Denmark

Corresponding authors
Geneviève Richard,
genevieve.richard@medisin.uio.no
Lars T. Westlye,
l.t.westlye@psykologi.uio.no

## ABSTRACT

Multimodal imaging enables sensitive measures of the architecture and integrity of the human brain, but the high-dimensional nature of advanced brain imaging features poses inherent challenges for the analyses and interpretations. Multivariate age prediction reduces the dimensionality to one biologically informative summary measure with potential for assessing deviations from normal lifespan trajectories. A number of studies documented remarkably accurate age prediction, but the differential age trajectories and the cognitive sensitivity of distinct brain tissue classes have yet to be adequately characterized. Exploring differential brain age models driven by tissue-specific classifiers provides a hitherto unexplored opportunity to disentangle independent sources of heterogeneity in brain biology. We trained machine-learning models to estimate brain age using various combinations of FreeSurfer based morphometry and diffusion tensor imaging based indices of white matter microstructure in 612 healthy controls aged 18–87 years. To compare the tissue-specific brain ages and their cognitive sensitivity, we applied each of the 11 models in an independent and cognitively well-characterized sample ($n = 265$, 20–88 years). Correlations between true and estimated age and mean absolute error (MAE) in our test sample were highest for the most comprehensive brain morphometry ($r = 0.83$, CI:0.78–0.86, MAE = 6.76 years) and white matter microstructure ($r = 0.79$, CI:0.74–0.83, MAE = 7.28 years) models, confirming sensitivity and generalizability. The deviance from the chronological age were sensitive to performance on several cognitive tests for various models, including spatial Stroop and symbol coding, indicating poorer performance in individuals with an over-estimated age. Tissue-specific brain age models provide sensitive measures of brain integrity, with implications for the study of a range of brain disorders.

Subjects Neuroscience, Psychiatry and Psychology, Radiology and Medical Imaging, Data Science
Keywords Machine learning, Brain age, Gray matter, White matter, DTI, T1

## INTRODUCTION

Increasing age is a major risk factor for cognitive decline and neurodegeneration, and deviating lifespan trajectories in brain structure and function is a sensitive marker in several common neurological and mental disorders (*Cole & Franke, 2017*). The maturing and aging brain is highly heterogeneous in term of individual trajectories and in term of brain regions and mechanisms involved (*Fjell et al., 2013*; *Westlye et al., 2010b*). Understanding the individual determinants and heterogeneity of the developing and aging brain is imperative for identifying persons at risk for various brain disorders, and for developing and applying effective and targeted treatments.

Exploring different modalities acquired by magnetic resonance imaging (MRI) provides a powerful tool to investigate age-related differences in both gray- and white-matter tissue classes across brain regions. However, the richness and complexity of the information provided by advanced imaging pipelines challenges its interpretation. Together, the multifactorial age-related variability and the richness of imaging measures have motivated the development of biologically informative summary measures based on brain imaging data. Using machine learning to estimate the biological age of the brain based on neuroimaging data is one such approach (*Cole & Franke, 2017*; *Cole et al., 2018b*; *Kaufmann et al., 2018*). Deviation from the normative trajectory is a highly relevant biomarker for the integrity of the brain in healthy and clinical populations (*Marquand et al., 2016*; *Wolfers et al., 2018*). Brain age gap is a heritable trait showing regionally specific genetic overlaps with major brain disorders, including schizophrenia and multiple sclerosis (*Kaufmann et al., 2018*), and accumulating evidence supports increased brain age in several clinical groups, including patients with schizophrenia (*Kaufmann et al., 2018*; *Schnack et al., 2016*), Alzheimer's disease (*Amoroso et al., 2017*; *Kaufmann et al., 2018*), HIV (*Cole et al., 2017*; *Kuhn et al., 2018*), multiple sclerosis (*Kaufmann et al., 2018*), and cardiovascular risk factors (*Franke et al., 2013*; *Habes et al., 2016*). Indeed, while individuals with brains estimated as younger than their chronological age have been shown to be more physically active (*Steffener et al., 2016*), augmented brain age has been associated with poor health (*Ronan et al., 2016*), poor cognitive performance (*Liem et al., 2017*), early neurodegenerative diseases (*Gaser et al., 2013*), and increased mortality (*Cole et al., 2018a*). Less is known about the biological and regional heterogeneity, i.e., to which degree different brain regions, systems or compartments show differential aging patterns and sensitivity to cognitive performance. Brain gray and white matter compartments, which can be assessed and quantified using T1-weighted imaging and diffusion tensor imaging (DTI), respectively, comprise distinct tissue classes with largely differential biological and environmental modifiers and age trajectories (*Bennett et al., 2010*; *Cao et al., 2017*; *Fjell et al., 2013*; *Salat et al., 2005*; *Storsve et al., 2014*; *Westlye et al., 2010a*; *Westlye et al., 2010b*). Therefore, allowing for differential brain age models for these distinct classes provides an opportunity to disentangle independent sources of heterogeneity in brain aging.

Thus, to identify common and unique aging patterns with potentially differential sensitivity to cognitive function, we aimed to test the complementary value of tissue-specific prediction by comparing brain age estimated using different combinations of FreeSurfer

based morphometric measures (regional cortical thickness, surface area and volume) and white matter microstructure (DTI based fractional anisotropy and mean, radial and axial diffusivity) across the brain. Based on previous studies on brain aging, we expected high accuracy and generalizability of the age prediction models (*Cole & Franke, 2017*). Since tissue specific brain age models capture biologically distinct information, we anticipated that the different FreeSurfer based brain morphometry and white matter microstructure models would only partly reflect common variance, and therefore provide complementary information with differential sensitivity to cognitive performance. Given that brain age predictions might be sensitive to the overall integrity of the brain (*Liem et al., 2017*), we anticipated that adult individuals in the targeted age range who show and over-estimated brain age would also show lower cognitive performance, in particular among the elderly, and that the tissue-specific brain age models would show partly differential cognitive sensitivity.

To ensure generalizability, we trained the models in a large publicly available training set ($n = 612$, 18–87 years) and validated their performance using 10-fold cross-validation before applying to an independent and well characterized test set ($n = 265$, 20–88 years). We assessed the cognitive sensitivity using linear and non-linear models with performance on a range of paper-and-pencil and computerized tests comprising different large-scale cognitive domains (processing speed, executive functioning, working memory, attention, and general intellectual abilities) and cognitive scores based on computational models as dependent variables and age, sex and brain age gap (BAG, estimated brain age minus chronological age) as independent variables. For transparency, we report results both at an uncorrected level and corrected using false discovery rate (FDR) and Bonferroni methods to control the error rate.

## MATERIAL AND METHODS

Figure 1 displays a flowchart of the main analysis pipeline. Table 1 summarizes key demographics. We included data from healthy volunteers from two independent cohorts: (1) the Cambridge Centre for Ageing and Neuroscience (Cam-CAN) sample (http://www.mrc-cbu.cam.ac.uk/datasets/camcan/; *Shafto et al., 2014*; *Taylor et al., 2017*) and (2) StrokeMRI, which is an ongoing study on the determinants of stroke recovery, brain health and successful aging (*Dorum et al., 2016*; *Dorum et al., 2017*). Figure 2 shows the age distribution for each sample. The distribution of age ($t = -2.09$, $p = 0.037$) and sex ($\chi^2(1) = 10.92$, $p < 0.001$) differed between samples.

Volunteers were recruited to Cam-CAN through a large-scale collaborative research project funded by the Biotechnology and Biological Sciences Research Council (BBSRC, grant number BB/H008217/1), the UK Medical Research Council and University of Cambridge. For more information, see http://www.cam-can.org. Among the 650 datasets made available, 17 were excluded based on missing or poor quality DTI data and 21 due to poor T1-weighted data quality. Data from the remaining 612 individuals (age 18–87, mean = 54.41, SD = 18.26, 314 females) were included.

Healthy individuals were recruited to StrokeMRI through advertisement in newspapers, social media and word-of-mouth. All participants completed a comprehensive cognitive

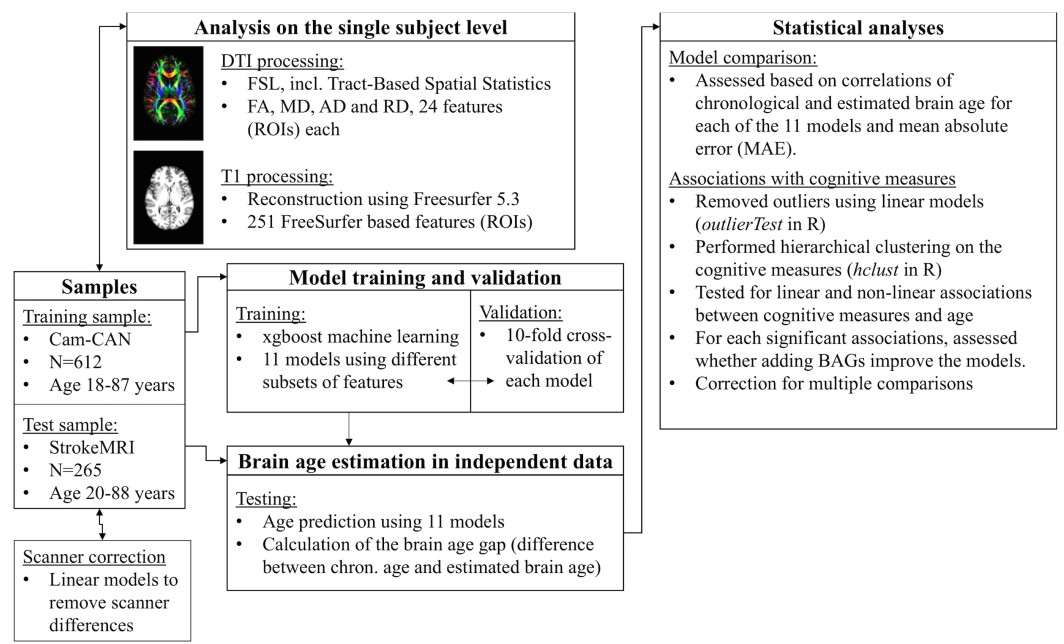

**Figure 1** Flowchart of the main analysis pipeline.

assessment, multimodal MRI and blood sampling for clinical biochemical analysis, various biomarkers and genotyping. MRI and cognitive assessments were performed on two subsequent days. Exclusion criteria included history of stroke, dementia, or other neurologic and psychiatric diseases, alcohol- and substance abuse, medications significantly affecting the nervous system and counter indications for MRI. In addition, individuals scoring lower than 25 on the Montreal Cognitive Assessment (MoCA; *Nasreddine et al., 2005*) were assessed for inclusion based on their age, level of education and performance on other cognitive tests. No participants were excluded based on a single low score. A neuroradiologist reviewed all scans and 14 participants with clinically significant abnormalities were excluded. Additional exclusion criteria included missing or incomplete MRI or cognitive data ($n = 2$), or poor quality images ($n = 20$). The remaining 265 participants (age 20–88, mean = 56.95, SD = 14.84, 168 females) were included in further analyses. The study was approved by the Regional Committee for Medical and Health Research Ethics (South-East Norway, REK 2014/694), and conducted in accordance with the Helsinki declaration. All subjects signed an informed consent prior to participating and received a compensation for their participation.

## Cognitive assessment in StrokeMRI

Cognitive performance was assessed with a set of neuropsychological and computerized tests assumed to be sensitive to cognitive aging, including the MoCA, the vocabulary and matrix subtests of the Wechsler Abbreviated Scale of Intelligence (WASI; *Wechsler, 1999*), the California Verbal Learning Test (CVLT-II; *Delis et al., 2000*), and the Delis-Kaplan Executive Function System (D-KEFS) color word interference test (Stroop; *Delis,*

**Table 1 Demographics and cognitive information.**

| | Cam-CAN | StrokeMRI Mean (SD) | Range (IQR) | Main effect age t (p) | Main effect sex t (p) |
|---|---|---|---|---|---|
| Total N (% females) | 612 (51.3%) | 265 (63.4%) | | | |
| Mean age (SD) | 54.41 (18.26) | 56.95 (14.84) | | | |
| Age range | 18–87 | 20–88 | | | |
| MoCA | – | 27.60 (1.72) | 21–30 (2) | −4.57 (<0.001)** | −2.32 (0.021) |
| WASI words | – | 65.27 (6.60) | 44–79 (10) | 4.72 (<0.001)** | 0.10 (0.920) |
| WASI matrix | – | 25.39 (5.64) | 7–35 (6) | −7.60 (<0.001)** | −0.28 (0.776) |
| CVLT learning 1-5 | – | 48.92 (11.37) | 17–73 (15.5) | −5.05 (<0.001)** | −5.26 (<0.001) |
| CVLT interference | – | 5.53 (2.15) | 0–13 (3) | −4.33 (<0.001)** | −0.41 (0.681) |
| CVLT recall | – | 10.83 (3.42) | 0–16 (5) | −6.50 (<0.001)** | 5.94 (<0.001) |
| CVLT delayed recall | – | 11.39 (3.44) | 0–16 (5) | −4.97 (<0.001)** | −5.51 (<0.001) |
| CVLT recognition hit | – | 14.70 (1.50) | 8–16 (2) | −2.62 (0.0093)* | −2.68 (0.008) |
| CVLT recognition errors | – | 3.79 (3.92) | 0–18 (4) | 5.22 (<0.001)** | 4.18 (<0.001) |
| CVLT recog misses | – | 1.30 (1.49) | 0–8 (2) | 2.62 (0.0093)* | 2.68 (0.008) |
| CVLT recog false alarm | – | 2.46 (3.48) | 0–18 (3) | 4.45 (<0.001)** | 3.59 (0.0004) |
| CVLT recog correct rejection | – | 44.20 (3.92) | 30–48 (4) | −5.22 (<0.001)** | −4.18 (<0.001) |
| CVLT d′ | – | 2.97 (0.72) | 0.97–3.90 (1.11) | −5.01 (<0.001)** | −4.50 (<0.001) |
| STROOP 1 | – | 31.14 (5.66) | 21–50 (7) | 5.05 (<0.001)** | 2.44 (0.015) |
| STROOP 2 | – | 22.12 (3.49) | 14–35 (4) | 2.89 (0.004)* | 2.27 (0.024) |
| STROOP 3 | – | 55.86 (14.13) | 10–108 (15) | 7.55 (<0.001)** | 2.97 (0.003) |
| STROOP 4 | – | 61.74 (14.85) | 33–117 (19) | 7.51 (<0.001)** | 1.77 (0.078) |
| STROOP mean 1 and 2 | – | 26.54 (4.16) | 18.5–42 (5) | 4.47 (<0.001)** | 2.47 (0.014) |
| STROOP 3 minus mean 1 and 2 | – | 81.94 (16.51) | 34.5–145 (18.5) | 7.31 (<0.001)** | 3.02 (0.003) |
| STROOP 4 minus mean 1 and 2 | – | 87.64 (16.73) | 53.5–142 (24) | 7.52 (<0.001)** | 1.85 (0.066) |
| CP—Right motor speed | – | 79.56 (23.34) | 34–153 (32) | −12.25 (<0.001)** | −0.36 (0.716) |
| CP—Left motor speed | – | 81.36 (17.80) | 39–131 (26) | −12.07 (<0.001)** | 0.20 (0.842) |
| CP—FAS Phonological flow | – | 54.70 (14.53) | 14–95 (19.75) | −0.61 (0.541) | −2.58 (0.011) |
| CP—FAS Semantic flow | – | 51.00 (10.14) | 27–81 (13) | −2.93 (0.004)* | −3.93 (<0.001) |
| CP—Visual WM forward ls | – | 4.23 (1.01) | 2–7 (2) | −5.31 (<0.001)** | 0.29 (0.774) |
| CP—Visual WM forward ss | – | 5.45 (1.87) | 1–10 (3) | −6.59 (<0.001)** | −0.25 (0.803) |
| CP—Visual WM backward ls | – | 3.80 (1.28) | 0–8 (1) | −4.60 (<0.001)** | −1.85 (0.065) |
| CP—Visual WM backward ss | – | 4.56 (2.08) | 0–12 (3) | −5.48 (<0.001)** | −1.02 (0.309) |
| CP—Visual WM ss | – | 9.96 (3.57) | 1–21 (4) | −7.04 (<0.001)** | −0.95 (0.342) |
| CP—Spatial stroop congruent (ms) | – | 674.42 (132.77) | 410–1159 (181) | 8.52 (<0.001)** | −1.03 (0.304) |
| CP—Spatial stroop incongruent (ms) | – | 929.52 (198.01) | 462–1827 (269) | 9.41 (<0.001)** | −0.75 (0.451) |
| CP—Spatial stroop Errors | – | 2.17 (2.41) | 0–11 (3) | 0.73 (0.463) | 1.59 (0.113) |
| CP—Spatial stroop numb of reps | – | 119.63 (16.64) | 55–166 (22) | −9.67 (<0.001)** | 1.23 (0.219) |
| CP—Spatial stroop incong–cong (ms) | – | 252 (110) | 20–678 (134.5) | 5.73 (<0.001)** | −0.68 (0.498) |
| CP—Spatspan ls | – | 5.37 (1.78) | 1–10 (2) | −9.12 (<0.001)** | −4.88 (<0.001) |
| CP—Spatspan tot | – | 29.87 (12.43) | 3–55 (18) | −9.28 (<0.001)** | −4.66 (<0.001) |

**Table 1** (*continued*)

|  | Cam-CAN | StrokeMRI Mean (SD) | Range (IQR) | Main effect age *t (p)* | Main effect sex *t (p)* |
|---|---|---|---|---|---|
| CP—Coding corr | – | 54.50 (12.11) | 24–88 (16) | −16.69 (<0.001)[**] | −2.46 (0.015) |
| CP—Coding error | – | 0.67 (0.99) | 0–5 (1) | −1.10 (0.271) | 1.56 (0.121) |
| TVA—Short-term memory storage ($K$) | – | 3.38 (0.77) | 1.46–5.53 (1.09) | −7.75 (<0.001)[**] | −1.52 (0.129) |
| TVA—Processing speed ($C$) | – | 31.55 (14.07) | 5.99–89.67 (14.75) | −4.69 (<0.001)[**] | 0.41 (0.6847) |
| TVA—Perceptual threshold ($t_0$) | – | 23.01 (14.05) | 0–79.75 (17.59) | 5.72 (<0.001)[**] | −1.94 (0.053) |
| TVA—Error rate | – | 0.10 (0.06) | 0.0035–0.3316 (0.0983) | −1.35 (0.177) | 0.67 (0.502) |
| Cluster 1 | – | – | – | −7.19 (<0.001)[**] | −5.16 (<0.001) |
| Cluster 2 | – | – | – | −7.28 (<0.001)[**] | 1.61 (0.110) |
| Cluster 3 | – | – | – | −2.01 (0.045)[*] | −3.99 (<0.001) |
| Cluster 4 | – | – | – | −9.98 (<0.001)[**] | 1.25 (0.212) |
| Cluster 5 | – | – | – | −6.86 (<0.001)[**] | −2.56 (0.011) |
| Cluster 6 | – | – | – | −15.79 (<0.001)[**] | −1.08 (0.282) |
| Cluster 7 | – | – | – | −6.50 (<0.001)[**] | −0.77 (0.440) |

**Notes.**
[*]Significant associations between cognitive measures with age after FDR correction.
[**]Significant associations between cognitive measures with age after Bonferroni correction.
IQR, interquartile range; MoCA, Montreal Cognitive Assessment; WASI, Wechsler Abbreviated Scale of Intelligence; CVLT, California Verbal Learning Test; STROOP, Delis-Kaplan Executive Function System (D-KEFS) color word interference test; CP, CabPad; WM, working memory; TVA, Theory of Visual Attention; ls, longest serie; ss, sum scores; tot, total.

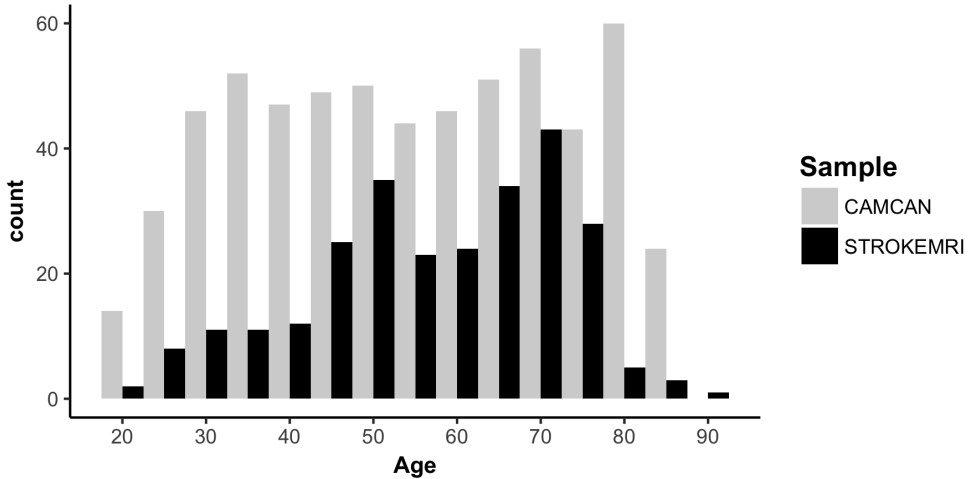

**Figure 2** Histogram of the age distribution for each sample.

*Kaplan & Kramer, 2001*). We included several computerized tests from the Cognitive Assessment at Bedside for iPAD (CABPad; *Willer et al., 2016*), including motor speed, verbal fluency (phonological and semantic), working memory, spatial Stroop (executive control of attention), spatial attention span, and symbol digit coding tests. In addition, in order to assess the specificity of cognitive associations using computation modeling, we included three mathematically independent parameters based on the Theory of Visual Attention (TVA; *Bundesen, 1990*; *Bundesen & Habekost, 2008*), including short-term memory storage ($K$), processing speed ($C$), perceptual threshold ($t_0$). These parameters

based on computational modeling of response patterns have been shown to be sensitive to age, brain structure and function in healthy individuals (*Espeseth et al., 2014*; *Wiegand et al., 2018*) and a range of brain disorders (*Habekost, 2015*; *Habekost & Starrfelt, 2009*). Here, we used a TVA-based modeling of a whole report (*Sperling, 1960*), in which six letters were briefly presented for different exposure durations and the participant's task was to accurately report as many letters as possible. Task error rate was also assessed (i.e., number of incorrect letters out of reported letters).

## MRI acquisition

Cam-CAN participants were scanned on a 3T Siemens TIM Trio scanner with a 32-channel head-coil at Medical Research Council (UK) Cognition and Brain Sciences Unit (MRC-CBSU) in Cambridge, UK. DTI data was acquired using a *twice—refocused* spin echo sequence with the following parameters a repetition time (TR) of 9,100 ms, echo time (TE) of 104 ms, field of view (FOV) of $192 \times 192$ mm, voxel size: 2 mm$^3$, 66 axial slices using 30 directions with $b = 1000$ s/mm$^2$, 30 directions with $b = 2000$ s/mm$^2$, and 3 $b = 0$ images (*Shafto et al., 2014*). Only the $b = [0, 1000]$ were used in the current analysis. High-resolution 3D T1-weighted data was acquired using a magnetization prepared rapid gradient echo (MPRAGE) sequence with the following parameters: TR: 2,250 ms, TE: 2.99 ms, inversion time (TI): 900 ms, flip angle: 9°, FOV of $256 \times 240 \times 192$ mm; voxel size = 1 mm$^3$ isotropic, GRAPPA acceleration factor of 2, scan time 4:32 min (*Shafto et al., 2014*).

StrokeMRI participants were scanned on a 3T GE 750 Discovery MRI scanner with a 32-channel head coil at Oslo University Hospital. Paddings were used to reduce head motion. DTI data were acquired using an echo planar imaging (EPI) sequence with the following parameters: TR/TE/flip angle: 8,150 ms/83.1 ms/90°, FOV: $256 \times 256$ mm, slice thickness: 2 mm, in-plane resolution: 2 mm, 60 directions ($b = 1000$ s/mm$^2$) and 5 $b = 0$ volumes, scan time: 8:58 min. In addition, 7 $b = 0$ volumes with reversed phase-encoding direction were acquired. High-resolution T1-weighted data was acquired using a 3D IR-prepared FSPGR (BRAVO) with the following parameters: TR: 8.16 ms, TE: 3.18 ms, flip angle: 12°, voxel size: $1 \times 1 \times 1$ mm, FOV: $256 \times 256$ mm, 188 sagittal slices, scan time: 4:43 min.

## DTI processing and analysis

Diffusion MRI data from both samples were processed locally using the Oxford Center for Functional Magnetic Resonance Imaging of the Brain (FMRIB) Software Library (FSL) (http://www.fmrib.ox.ac.uk/fsl). To correct for geometrical distortions, motion and eddy currents, data were preprocessed using topup (https://fsl.fmrib.ox.ac.uk/fsl/fslwiki/topup) and eddy (https://fsl.fmrib.ox.ac.uk/fsl/fslwiki/eddy) respectively (*Andersson, Skare & Ashburner, 2003*; *Smith et al., 2004*). Topup uses information from the reversed phase-encoded image, resulting in pairs of images (blip-up, blip-down) with distortions going in opposite directions. From these image pairs the susceptibility-induced off-resonance field was estimated and the two images were combined into a single corrected one (*Andersson, Skare & Ashburner, 2003*; *Smith et al., 2004*). This step was performed on StrokeMRI data only. Eddy detects and replaces slices affected by signal loss due to bulk motion during

diffusion encoding, which is performed within an integrated framework along with correction for susceptibility induced distortions, eddy currents and motion (*Andersson & Sotiropoulos, 2016*). Although these processing steps have been shown to strongly increase the temporal signal-to-noise ratio (tSNR) (*Doan et al., 2017*), we performed additional visual inspection to identify and remove poor quality data, such as data that failed the processing steps due to low quality.

Fractional anisotropy (FA), eigenvector, and eigenvalue maps were calculated using dtifit in FSL. Mean diffusivity (MD) was defined as the mean of all three eigenvalues, radial diffusivity (RD) as the mean of the second and third eigenvalue, and axial diffusivity (AD) as the principal eigenvalue.

Voxelwise analysis of FA, MD, AD and RD were carried out using Tract-Based Spatial Statistics (TBSS; *Smith et al., 2006*), part of FSL (*Smith et al., 2004*). First, all subjects' FA data were aligned to a common space using the nonlinear registration tool FNIRT (*Andersson, Jenkinson & Smith, 2007a*; *Andersson, Jenkinson & Smith, 2007b*). Next, the mean FA image was created and thinned to create a mean FA skeleton, which represents the centers of all tracts common to all participants. Each subject's aligned FA data was then projected onto this skeleton and the resulting data fed into voxelwise cross-subject statistics. The same warping and skeletonization was repeated for MD, AD and RD. We thresholded and binarized the mean FA skeleton at FA > 0.2. For each individual, we calculated the mean skeleton FA, MD, AD and RD, as well as mean values within 23 regions of interest (ROIs) based on two probabilistic white matter atlases provided with FSL, i.e., the CBM-DTI-81 white-matter labels atlas and the JHU white-matter tractography atlas (*Hua et al., 2008*; *Mori et al., 2005*; *Wakana et al., 2007*), yielding a total of 96 DTI features per individual.

## T1 processing

T1-weighted images from both samples were processed using FreeSurfer 5.3 (http://surfer.nmr.mgh.harvard.edu; *Dale, Fischl & Sereno, 1999*) including brain extraction, intensity normalization, automated tissue segmentation, generation of white and pial surfaces (*Dale, Fischl & Sereno, 1999*). All reconstructions were visually assessed and edited by trained research personnel where appropriate. The reconstructions that failed the corrections were excluded from further analysis, such as data with excessive movement artefacts. Cortical parcellation was performed using the Desikan–Killiany atlas (*Desikan et al., 2006*; *Fischl et al., 2004*) and subcortical segmentation was performed based on a probabilistic atlas (*Fischl et al., 2002*). In addition to global features (intracranial volume, total surface area, whole-cortex mean thickness), mean thickness, total surface area, and volume for each cortical ROI, as well as the volume of subcortical structures were computed yielding a set of 251 FreeSurfer based features.

## Age prediction

Eleven different models were trained to estimate age based on the feature sets described above (one based on FreeSurfer T1 features, one based on WM DTI features, one including all T1 and DTI features, in addition to eight models based on a smaller subset of features,

including models based on FA, MD, AD, RD, sub-cortical volume, volume, area and thickness to further explore the modality specificity of the estimations).

Due to systematic differences in brain features between scanners (*Madan, 2017*) as well as non-linear effects of age, we regressed out main effects of scanner using linear models including age, age squared, sex and scanner for each feature, and used the fitted data in further analysis for brain age prediction. In addition, we regressed out the estimated total intracranial volume from the area and volume features. Next, for each model, we created a training data matrix by concatenating all the features for all participants in the training sample (Cam-CAN), which were used as input to estimate age. We used the *xgboost* framework in R (http://xgboost.readthedocs.io/en/latest/R-package/xgboostPresentation.html), an efficient and scalable implementation of gradient boosting machine learning techniques, to build the prediction models. The following parameters were used: learning rate (eta) = 0.1, nround = 5,000, gamma = 1, max_depth = 6, subsample = 0.5. To estimate the performance of our models, we used a 10-fold cross-validation procedure within the training sample and repeated the cross-validation step 1,000 times to provide a robust estimate of model predictive accuracy. Next, we tested the performance of our trained models by predicting age in unseen healthy subjects in the test sample (StrokeMRI).

For each feature set, we calculated the correlation between the predicted and the chronological age as a measure of the model performance, in addition to the mean absolute error (MAE, in years). For each individual, we calculated the discrepancy between the estimated and the chronological age, i.e., the BAG, for each model. The MAE was calculated from the BAG for each model. Since we were interested in the effect of BAG independently of age, the effect of age was regressed out for each BAG using linear models.

## Statistical analysis

Statistical analysis was performed using R (*R Core Team, 2017*). For cognitive data, we used *outlierTest* from the car package (*Fox & Weisberg, 2011*) to identify the most extreme observations based on a linear model, including age and sex. Twenty-five observations were identified as outliers and treated as missing values based on a Bonferroni corrected $p < 0.05$. To visualize the associations between the cognitive tests and to form cognitive domain scores based on the correlation patterns, we performed hierarchical clustering using the default setting of the heatmap.2 package in gplots (*Warnes et al., 2016*), which uses hclust (*Müllner, 2013*) to form clusters based on the complete linkage method. Briefly, this is a step-wise clustering process that merges the two nearest clusters until only one single cluster remains, maximizing distance between individuals components between two clusters.

For each cognitive measure and summary score based on the clusters formed form the clustering step above, we used linear models to test for the effect of age and sex. Since cognitive performance may show non-linear associations with age, we performed an additional analysis including both age and age$^2$ in the models. Then, for each test showing a significant association with age, we tested whether adding BAG to the models lead to an improved model fit. More specifically, we tested for differential associations with

cognitive function by comparing the parameter estimates for the different BAG models using Fisher $z$-transformation. To test the assumption that increased BAG is more relevant for cognitive function among the elderly, we tested for age by BAG interactions on cognitive performance. For transparency, we report both uncorrected $p$-values and $p$-values adjusted using FDR (*Benjamini & Hochberg, 1995*; *Wright, 1992*) and Bonferroni correction using a factor of 495 (11 brain gaps and 45 cognitive features).

## RESULTS

### Brain age prediction

Ten-fold cross-validation on the training sample (Cam-CAN) revealed high correlations between chronological and predicted age for the DTI based white matter microstructure ($r = 0.87$) and FreeSurfer based morphometric ($r = 0.88$) models. Likewise, the correlations for FA ($r = .76$), MD ($r = .80$), AD ($r = .83$), RD ($r = 78$), sub-volume ($r = .84$), volume ($r = .80$), area ($r = .70$) and thickness ($r = .79$) based models also confirmed reasonable model performance.

Most models accurately predicted age in the independent test set (StrokeMRI). Figure 3A shows a correlation matrix for the 11 BAGs. Figure 3B shows the correlations between the chronological age and the predicted age in the test sample for each model with their confidence intervals, ranging from ($r = .86$, CI:.82–.89, MAE $= 6.14$) for the combined model to $r = .58$ (CI:.49–.65, MAE $= 10.24$) for the model based on area. Figure 3C is described below. Figures 3D to 3F show the estimated age from the three models that performed best among the 11 feature sets, i.e., the combined DTI and T1 feature models ($r = .86$, CI:.82–.89, MAE $= 6.14$), the 251 FS T1 features ($r = .83$, CI:.78–.86, MAE $= 6.76$), and the 96 WM DTI features ($r = .79$, CI:.74–.83, MAE $= 7.28$).

### Cognitive assessments and associations with BAGs in StrokeMRI

Figure 4 shows a correlation matrix across all normalized cognitive scores with the variables sorted according to the hierarchical clustering used in the main analysis. Several variables were highly correlated, and the clustering solution generally suggested seven broad cognitive domains including (Cluster 1) memory and learning (CVLT, attention span, MoCA), (Cluster 2) visual processing speed (TVA processing speed and perceptual threshold), (Cluster 3) verbal skills (phonological and semantic flow), (Cluster 4) attentional control and speed (spatial Stroop), (Cluster 5) executive control and speed (color-word Stroop), (Cluster 6) reasoning and psychomotor speed (matrix, symbol coding and motor speed, short-term memory storage (TVA-parameter $K$)), and (Cluster 7) working memory. Table 1 summarizes descriptive statistics and associations with age and sex for each of the 49 cognitive scores, derived features and domain scores. Linear models revealed 45 significant associations with age after correcting for multiple comparisons, with the strongest effect sizes for the symbol coding test, motor speed, spatial Stroop and spatial attention span. Since non-linear models revealed significant associations with age$^2$ only with the color word Stroop 3 (inhibition) and its derived scores (See Table S1), the main models presented here are linear in order to keep the model to its simplest form.

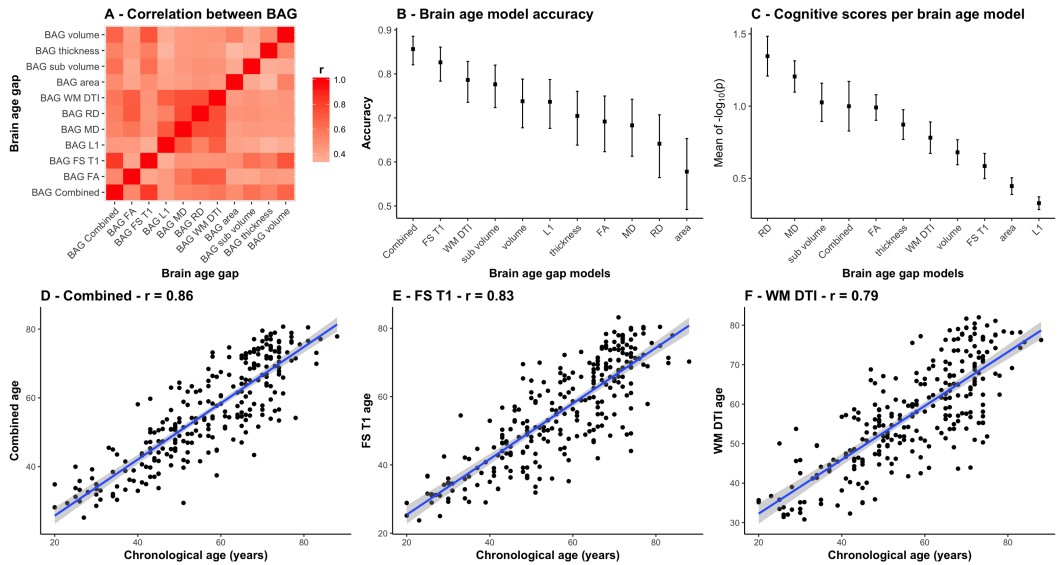

**Figure 3   Comparison between the 11 BAG models.** (A) Heatmap of the correlation between different BAGs. (B) Correlations between the chronological age and the predicted age in the test sample for each model with their confidence intervals. (C) Mean and standard error of the 45 $p$-values ($-\log_{10}(p)$) for the cognitive scores and composite scores for each row (i.e., BAGs), with a higher mean representing a stronger global association across tests. (D) Correlation between the chronological age of each subjects and the combined age, (E) the brain morphometry age, and (F) the white matter microstructure age.

Table 2 shows summary statistics for the associations between cognitive performance and BAG using linear models, including age and sex as covariates. Figure 5 shows a heatmap of the association between cognitive scores and brain age gaps for which the significant associations have been marked with an asterisk. Table S1 and Fig. S1 shows the summary statistics and the heatmap of the associations between cognitive performance and BAG using non-linear models. Figure 3C shows the mean and standard error of the 45 $p$-values ($-\log_{10}(p)$) for the cognitive scores and composite scores for each row (i.e., BAGs), with a higher mean representing a stronger cumulative association across tests.

Figure 6 shows a scatter plot of the two strongest associations, which were found between the most comprehensive model (all features combined) and spatial Stroop congruent trials and number of responses, respectively, indicating poorer performance with higher BAG. Fisher z-transformation revealed no statistically significant differences in the cognitive associations between linear models using tissue-specific BAG. No significant interactions were found between BAG and age on cognitive performance.

## DISCUSSION

Brain aging is highly heterogeneous, and expanding our understanding of the biological determinants of human aging is imperative for reducing the burden of age-related cognitive decline and neurodegenerative disorders. An estimate of an individual's deviation from the expected lifespan trajectory in brain structure and function may provide a sensitive

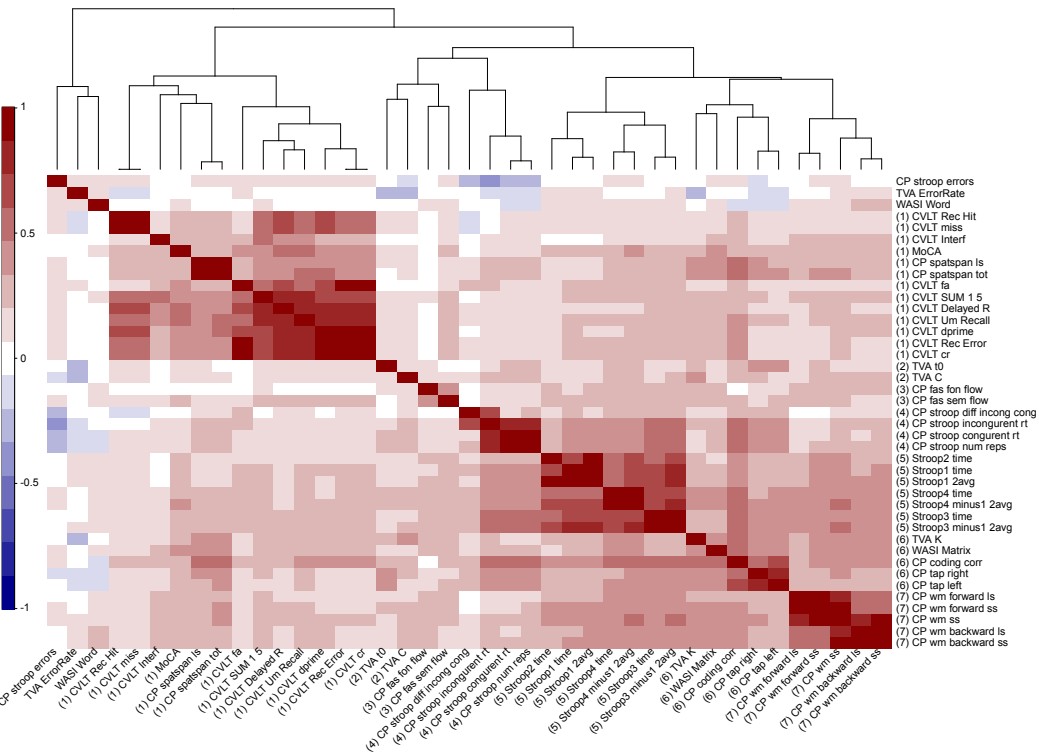

**Figure 4** **Hierarchical clustering of the cognitive features.** Each cognitive score was normalized and when required the scores were multiplied by −1 to ensure that positive scores represent good performance. The higher panel shows the dendrogram resulting from the hierarchical clustering of the scores in seven cognitive domains. Table S2 provides detailed overview of all abbreviations used.

measure of individual brain integrity and health, both in presumably healthy individuals and in patients suffering from various brain disorders.

The biological heterogeneity of the brain strongly suggests that the concept of a single brain age is too simple, and that tissue-specific brain age models may provide increased sensitivity and specificity in relation to cognitive and mental functions. In line with this view, our main findings demonstrate that different combinations of FreeSurfer based brain morphometry and DTI based white matter microstructural indices can be used to accurately predict the age of individuals, but that the shared variance from the different models suggest that they reflect partly non-overlapping processes of brain aging. Further, the results revealed partly differential sensitivity to cognitive performance; with the strongest cumulative associations across cognitive tests for brain age gaps estimated using RD. Even though our data provide no strong evidence of independent associations with cognitive performance in the current sample of healthy individuals, tissue specific age prediction models might better inform us about the individual determinants and heterogeneity of the aging brain compared to models collapsing several brain compartments by potentially capturing distinct measures of brain aging.

**Table 2  Cognitive associations with BAG—statistics.**

| Test | Adj R² no-BAG | BAG | Main effect age t (p) | Main effect sex t (p) | Main effect BAG t (p) | Adj R² |
|---|---|---|---|---|---|---|
| MoCA | 0.0907 | T1 | −4.5596 (<0.001) | −2.3145 (0.021) | −0.124 (0.901) | 0.0878 |
| | | DTI | −4.5599 (<0.001) | −2.3155 (0.021) | 1.5914 (0.113) | 0.0966 |
| | | Combined | −4.5653 (<0.001) | −2.3176 (0.021) | −0.4626 (0.644) | 0.0885 |
| WASI words | 0.0731 | T1 | 4.7118 (<0.001) | 0.1020 (0.919) | −0.2169 (0.828) | 0.0704 |
| | | DTI | 4.7056 (<0.001) | 0.1121 (0.911) | −0.8126 (0.417) | 0.0727 |
| | | Combined | 4.7091 (<0.001) | 0.1041 (0.917) | −0.4827 (0.630) | 0.0711 |
| WASI matrix | 0.1791 | T1 | −7.6061 (<0.001) | −0.2785 (0.781) | −0.9158 (0.361) | 0.1793 |
| | | DTI | −7.6610 (<0.001) | −0.2624 (0.793) | −1.6546 (0.099) | 0.1854 |
| | | Combined | −7.6128 (<0.001) | −0.2726 (0.785) | −1.1102 (0.268) | 0.1806 |
| CVLT learning 1-5 | 0.1810 | T1 | −5.0373 (<0.001) | −5.2514 (<0.001) | −0.2505 (0.802) | 0.1750 |
| | | DTI | −5.0418 (<0.001) | −5.2533 (<0.001) | −0.3608 (0.719) | 0.1753 |
| | | Combined | −5.0387 (<0.001) | −5.2522 (<0.001) | −0.2492 (0.803) | 0.1750 |
| CVLT interference | 0.0664 | T1 | −4.3256 (<0.001) | −0.4062 (0.685) | −0.9588 (0.339) | 0.0626 |
| | | DTI | −4.3218 (<0.001) | −0.4104 (0.682) | −0.2391 (0.811) | 0.0594 |
| | | Combined | −4.3202 (<0.001) | −0.4101 (0.682) | −0.1875 (0.851) | 0.0594 |
| CVLT recall | 0.2438 | T1 | −6.4897 (<0.001) | −5.9257 (<0.001) | −0.4868 (0.627) | 0.2397 |
| | | DTI | −6.4885 (<0.001) | −5.9257 (<0.001) | −0.1245 (0.901) | 0.2391 |
| | | Combined | −6.5080 (<0.001) | −5.9373 (<0.001) | −1.1114 (0.268) | 0.2427 |
| CVLT delayed recall | 0.1850 | T1 | −4.9636 (<0.001) | −5.4973 (<0.001) | 0.1421 (0.887) | 0.1808 |
| | | DTI | −4.9611 (<0.001) | −5.4969 (<0.001) | 0.224 (0.823) | 0.1809 |
| | | Combined | −4.9655 (<0.001) | −5.4954 (<0.001) | −0.3038 (0.762) | 0.1810 |
| CVLT recognition hits | 0.0494 | T1 | −2.6125 (0.010) | −2.6822 (0.008) | −0.8586 (0.391) | 0.0486 |
| | | DTI | −2.6144 (0.010) | −2.6786 (0.008) | 0.0946 (0.925) | 0.0459 |
| | | Combined | −2.6212 (0.009) | −2.6854 (0.008) | −1.0724 (0.285) | 0.0501 |
| CVLT recognition errors | 0.1526 | T1 | 5.2227 (<0.001) | 4.1850 (<0.001) | −0.8471 (0.398) | 0.1528 |
| | | DTI | 5.2115 (<0.001) | 4.1755 (<0.001) | −0.5651 (0.573) | 0.1514 |
| | | Combined | 5.2139 (<0.001) | 4.1740 (<0.001) | −0.2537 (0.800) | 0.1506 |
| CVLT recog misses | 0.0494 | T1 | 2.6125 (0.010) | 2.6822 (0.008) | 0.8586 (0.391) | 0.0486 |
| | | DTI | 2.6144 (0.010) | 2.6786 (0.008) | −0.0946 (0.925) | 0.0459 |
| | | Combined | 2.6212 (0.009) | 2.6854 (0.008) | 1.0724 (0.285) | 0.0501 |
| CVLT recog false alarm | 0.1150 | T1 | 4.4519 (<0.001) | 3.5827 (<0.001) | −0.776 (0.439) | 0.1146 |
| | | DTI | 4.4378 (<0.001) | 3.5803 (<0.001) | −0.5207 (0.603) | 0.1134 |
| | | Combined | 4.4418 (<0.001) | 3.5788 (<0.001) | −0.3488 (0.728) | 0.1129 |
| CVLT recog correct rejection | 0.1526 | T1 | −5.2227 (<0.001) | −4.1850 (<0.001) | 0.8471 (0.398) | 0.1528 |
| | | DTI | −5.2115 (<0.001) | −4.1755 (<0.001) | 0.5651 (0.573) | 0.1514 |
| | | Combined | −5.2139 (<0.001) | −4.1740 (<0.001) | 0.2537 (0.800) | 0.1506 |
| CVLT d' | 0.1566 | T1 | −5.0074 (<0.001) | −4.4914 (<0.001) | 0.3628 (0.717) | 0.1536 |
| | | DTI | −5.0021 (<0.001) | −4.4969 (<0.001) | 0.8538 (0.394) | 0.1556 |
| | | Combined | −5.0038 (<0.001) | −4.4902 (<0.001) | 0.1699 (0.865) | 0.1533 |
| STROOP 1 | 0.1118 | T1 | 5.1466 (<0.001) | 2.4999 (0.013) | 2.6939 (0.008) | 0.1299 |

**Table 2** (*continued*)

| Test | Adj R² no-BAG | BAG | Main effect age t (p) | Main effect sex t (p) | Main effect BAG t (p) | Adj R² |
|---|---|---|---|---|---|---|
| | | DTI | 5.0968 (<0.001) | 2.4769 (0.014) | 1.6664 (0.097) | 0.1147 |
| | | Combined | 5.2111 (<0.001) | 2.5317 (0.012) | 3.3767 (<0.001)* | 0.1434 |
| STROOP 2 | 0.0477 | T1 | 2.8868 (0.004) | 2.2619 (0.025) | 0.1557 (0.876) | 0.0433 |
| | | DTI | 2.8768 (0.004) | 2.2489 (0.025) | −0.4639 (0.643) | 0.0440 |
| | | Combined | 2.8949 (0.004) | 2.2713 (0.024) | 0.4976 (0.619) | 0.0442 |
| STROOP 3 | 0.2104 | T1 | 7.5930 (<0.001) | 2.9898 (0.003) | 1.5092 (0.133) | 0.2109 |
| | | DTI | 7.6511 (<0.001) | 3.0224 (0.003) | 2.231 (0.027) | 0.2190 |
| | | Combined | 7.6793 (<0.001) | 3.0233 (0.003) | 2.5768 (0.011) | 0.2240 |
| STROOP 4 | 0.1887 | T1 | 7.5403 (<0.001) | 1.7884 (0.075) | 1.2397 (0.216) | 0.1906 |
| | | DTI | 7.5847 (<0.001) | 1.8121 (0.071) | 1.7368 (0.084) | 0.1953 |
| | | Combined | 7.6387 (<0.001) | 1.8247 (0.069) | 2.3662 (0.019) | 0.2033 |
| STROOP mean 1 and 2 | 0.0949 | T1 | 4.5089 (<0.001) | 2.5033 (0.013) | 1.5875 (0.114) | 0.0978 |
| | | DTI | 4.4750 (<0.001) | 2.4760 (0.014) | 0.3927 (0.695) | 0.0894 |
| | | Combined | 4.5432 (<0.001) | 2.5399 (0.012) | 2.0254 (0.044) | 0.1034 |
| STROOP 3 minus mean 1 and 2 | 0.2051 | T1 | 7.3383 (<0.001) | 3.0427 (0.003) | 1.1397 (0.256) | 0.2021 |
| | | DTI | 7.3613 (<0.001) | 3.0703 (0.002) | 1.3546 (0.177) | 0.2038 |
| | | Combined | 7.4197 (<0.001) | 3.1063 (0.002) | 2.1881 (0.030) | 0.2130 |
| STROOP 4 minus mean 1 and 2 | 0.1936 | T1 | 7.5360 (<0.001) | 1.8671 (0.063) | 0.8763 (0.382) | 0.1919 |
| | | DTI | 7.5297 (<0.001) | 1.8697 (0.063) | 0.6331 (0.527) | 0.1907 |
| | | Combined | 7.6081 (<0.001) | 1.9215 (0.056) | 1.7531 (0.081) | 0.1993 |
| CP—Right motor speed | 0.3695 | T1 | −12.2893 (<0.001) | −0.3592 (0.720) | −1.5504 (0.122) | 0.3676 |
| | | DTI | −12.2318 (<0.001) | −0.3612 (0.718) | −0.3435 (0.732) | 0.3620 |
| | | Combined | −12.3125 (<0.001) | −0.3587 (0.720) | −1.8139 (0.071) | 0.3697 |
| CP—Left motor speed | 0.3630 | T1 | −12.1437 (<0.001) | 0.2100 (0.834) | −1.9945 (0.047) | 0.3634 |
| | | DTI | −12.0669 (<0.001) | 0.2081 (0.835) | −0.8704 (0.385) | 0.3555 |
| | | Combined | −12.2516 (<0.001) | 0.2149 (0.830) | −2.9047 (0.004) | 0.3740 |
| CP—FAS Semantic flow | 0.0840 | T1 | −2.9562 (0.003) | −3.9454 (<0.001) | −2.0826 (0.038) | 0.0960 |
| | | DTI | −2.9607 (0.003) | −3.9388 (<0.001) | −2.0997 (0.037) | 0.0963 |
| | | Combined | −2.9513 (0.004) | −3.9389 (<0.001) | −1.8308 (0.068) | 0.0926 |
| CP—Visual WM forward ls | 0.0936 | T1 | −5.3071 (<0.001) | 0.2850 (0.776) | −0.5838 (0.560) | 0.0906 |
| | | DTI | −5.3392 (<0.001) | 0.2963 (0.767) | −1.7204 (0.087) | 0.0999 |
| | | Combined | −5.3059 (<0.001) | 0.2853 (0.776) | −0.3127 (0.755) | 0.0897 |
| CP—Visual WM forward ss | 0.1416 | T1 | −6.5795 (<0.001) | −0.2502 (0.803) | −0.2158 (0.829) | 0.1375 |
| | | DTI | −6.6000 (<0.001) | −0.2448 (0.807) | −1.1695 (0.243) | 0.1420 |
| | | Combined | −6.5786 (<0.001) | −0.2496 (0.803) | −0.02 (0.984) | 0.1373 |
| CP—Visual WM backward ls | 0.0852 | T1 | −4.5941 (<0.001) | −1.8511 (0.065) | −0.1047 (0.917) | 0.0820 |
| | | DTI | −4.6170 (<0.001) | −1.8545 (0.065) | −1.3334 (0.184) | 0.0884 |
| | | Combined | −4.6051 (<0.001) | −1.8550 (0.065) | −0.8013 (0.424) | 0.0843 |
| CP—Visual WM backward ss | 0.1022 | T1 | −5.4741 (<0.001) | −1.0181 (0.310) | −0.2721 (0.786) | 0.1015 |
| | | DTI | −5.4971 (<0.001) | −1.0179 (0.310) | −1.3043 (0.193) | 0.1072 |
| | | Combined | −5.4898 (<0.001) | −1.0215 (0.308) | −1.0074 (0.315) | 0.1048 |

**Table 2** (*continued*)

| Test | Adj R² no-BAG | BAG | Main effect age t (p) | Main effect sex t (p) | Main effect BAG t (p) | Adj R² |
|---|---|---|---|---|---|---|
| CP—Visual WM ss | 0.1607 | T1 | −7.0322 (<0.001) | −0.9515 (0.342) | −0.3013 (0.763) | 0.1591 |
| | | DTI | −7.0622 (<0.001) | −0.9511 (0.342) | −1.3634 (0.174) | 0.1649 |
| | | Combined | −7.0399 (<0.001) | −0.9528 (0.342) | −0.6665 (0.506) | 0.1603 |
| CP—Spatial stroop congruent | 0.2288 | T1 | 8.6156 (<0.001) | −1.0080 (0.314) | 2.1921 (0.029) | 0.2288 |
| | | DTI | 8.6687 (<0.001) | −1.0021 (0.317) | 2.6995 (0.007) | 0.2362 |
| | | Combined | 8.8278 (<0.001) | −0.9828 (0.327) | 3.9007 (<0.001)[**] | 0.2588 |
| CP—Spatial stroop incongruent | 0.2548 | T1 | 9.5489 (<0.001) | −0.7429 (0.458) | 2.6569 (0.008) | 0.2700 |
| | | DTI | 9.5931 (<0.001) | −0.7587 (0.449) | 2.8817 (0.004) | 0.2735 |
| | | Combined | 9.7197 (<0.001) | −0.7378 (0.461) | 3.8071 (<0.001)[**] | 0.2903 |
| CP—Spatial stroop numb of reps | 0.2731 | T1 | −9.7755 (<0.001) | 1.2211 (0.223) | −2.2212 (0.027) | 0.2753 |
| | | DTI | −9.8507 (<0.001) | 1.2328 (0.219) | −2.9614 (0.003) | 0.2859 |
| | | Combined | −9.9891 (<0.001) | 1.2198 (0.224) | −3.8816 (<0.001)[**] | 0.3027 |
| CP—Spatial stroop incong–cong | 0.1012 | T1 | 5.7663 (<0.001) | −0.6595 (0.510) | 1.5611 (0.120) | 0.1134 |
| | | DTI | 5.7466 (<0.001) | −0.6678 (0.505) | 0.9705 (0.333) | 0.1081 |
| | | Combined | 5.7568 (<0.001) | −0.6584 (0.511) | 1.2056 (0.229) | 0.1099 |
| CP—Spatspan ls | 0.3055 | T1 | −9.1038 (<0.001) | −4.8656 (<0.001) | −0.032 (0.975) | 0.3009 |
| | | DTI | −9.1746 (<0.001) | −4.9104 (<0.001) | −1.5749 (0.117) | 0.3077 |
| | | Combined | −9.1043 (<0.001) | −4.8663 (<0.001) | −0.075 (0.940) | 0.3009 |
| CP—Spatspan total | 0.3057 | T1 | −9.2664 (<0.001) | −4.6439 (<0.001) | 0.1074 (0.915) | 0.3024 |
| | | DTI | −9.3260 (<0.001) | −4.6815 (<0.001) | −1.3773 (0.170) | 0.3076 |
| | | Combined | −9.2686 (<0.001) | −4.6461 (<0.001) | −0.0612 (0.951) | 0.3024 |
| CP—Coding corr | 0.5387 | T1 | −16.7647 (<0.001) | −2.5004 (0.013) | −1.6149 (0.108) | 0.5352 |
| | | DTI | −17.0893 (<0.001) | −2.5467 (0.012) | −3.3998 (<0.001)[*] | 0.5510 |
| | | Combined | −17.0071 (<0.001) | −2.5604 (0.011) | −3.0056 (0.003) | 0.5467 |
| TVA—Short-term memory storage (K) | 0.2013 | T1 | −7.7691 (<0.001) | −1.5196 (0.130) | −1.1179 (0.265) | 0.1981 |
| | | DTI | −7.8117 (<0.001) | −1.5383 (0.125) | −2.0302 (0.043) | 0.2070 |
| | | Combined | −7.7525 (<0.001) | −1.5195 (0.130) | −0.9537 (0.341) | 0.1970 |
| TVA—Perceptual threshold ($t_0$) | 0.0764 | T1 | 5.7303 (<0.001) | −1.9470 (0.053) | 0.9617 (0.337) | 0.1141 |
| | | DTI | 5.7333 (<0.001) | −1.9444 (0.053) | 1.1066 (0.270) | 0.1152 |
| | | Combined | 5.7523 (<0.001) | −1.9587 (0.051) | 1.8346 (0.068) | 0.1226 |
| TVA—Processing speed (C) | 0.1304 | T1 | −4.6692 (<0.001) | 0.3969 (0.692) | 0.8093 (0.419) | 0.0723 |
| | | DTI | −4.6800 (<0.001) | 0.4053 (0.686) | 0.1402 (0.889) | 0.0699 |
| | | Combined | −4.6827 (<0.001) | 0.3944 (0.694) | 0.8916 (0.374) | 0.0728 |
| Cluster 1 | 0.2470 | T1 | −7.1741 (<0.001) | −5.1567 (<0.001) | −0.1927 (0.847) | 0.2440 |
| | | DTI | −7.1623 (<0.001) | −5.1410 (<0.001) | 0.3683 (0.713) | 0.2443 |
| | | Combined | −7.1805 (<0.001) | −5.1641 (<0.001) | −0.3879 (0.699) | 0.2443 |
| Cluster 2 | 0.1720 | T1 | −7.2680 (<0.001) | 1.6030 (0.110) | −0.1013 (0.919) | 0.1687 |
| | | DTI | −7.2785 (<0.001) | 1.6062 (0.110) | −0.6549 (0.513) | 0.1701 |
| | | Combined | −7.2740 (<0.001) | 1.6104 (0.109) | −0.6382 (0.524) | 0.1700 |
| Cluster 3 | 0.0698 | T1 | −2.0177 (0.045) | −3.9824 (<0.001) | −0.8103 (0.419) | 0.0686 |
| | | DTI | −2.0337 (0.043) | −3.9969 (<0.001) | −1.84 (0.067) | 0.0783 |
| | | Combined | −2.0185 (0.045) | −3.9877 (<0.001) | −0.9765 (0.330) | 0.0697 |

**Table 2** (*continued*)

| Test | Adj R² no-BAG | BAG | Main effect age t (p) | Main effect sex t (p) | Main effect BAG t (p) | Adj R² |
|------|------|------|------|------|------|------|
| Cluster 4 | 0.2783 | T1 | −10.1319 (<0.001) | 1.2314 (0.219) | −2.5436 (0.012) | 0.2937 |
| | | DTI | −10.1479 (<0.001) | 1.2377 (0.217) | −2.5207 (0.012) | 0.2933 |
| | | Combined | −10.3013 (<0.001) | 1.2196 (0.224) | −3.6163 (<0.001)[*] | 0.3113 |
| Cluster 5 | 0.1772 | T1 | −6.8872 (<0.001) | −2.5902 (0.010) | −1.1084 (0.269) | 0.1779 |
| | | DTI | −6.8667 (<0.001) | −2.5805 (0.010) | −0.5825 (0.561) | 0.1750 |
| | | Combined | −6.9577 (<0.001) | −2.6481 (0.009) | −1.9103 (0.057) | 0.1858 |
| Cluster 6 | 0.5092 | T1 | −15.9345 (<0.001) | −1.1148 (0.266) | −1.8971 (0.059) | 0.5145 |
| | | DTI | −15.9719 (<0.001) | −1.1080 (0.269) | −2.0875 (0.038) | 0.5160 |
| | | Combined | −16.0156 (<0.001) | −1.1196 (0.264) | −2.459 (0.015) | 0.5193 |
| Cluster 7 | 0.1399 | T1 | −6.4852 (<0.001) | −0.7736 (0.440) | −0.3433 (0.732) | 0.1369 |
| | | DTI | −6.5210 (<0.001) | −0.7689 (0.443) | −1.6007 (0.111) | 0.1452 |
| | | Combined | −6.4926 (<0.001) | −0.7759 (0.439) | −0.63 (0.529) | 0.1379 |

**Notes.**
[*]FDR significant.
[**]Bonferroni significant.
MoCA, Montreal Cognitive Assessment; WASI, Wechsler Abbreviated Scale of Intelligence; CVLT, California Verbal Learning Test; STROOP, Delis-Kaplan Executive Function System (D-KEFS) color word interference test; CP, Cognitive Assessment at Bedside for iPAD (CabPAD); WM, working memory; TVA, Theory of Visual Attention; ls, longest serie; ss, sum scores; tot, total.

## Brain age prediction

For the age prediction models, our results demonstrated that the 11 different combinations of FreeSurfer based morphometric measures (regional cortical thickness, surface area and volume) and white matter microstructure features (diffusion tensor imaging (DTI) based fractional anisotropy and mean, radial and axial diffusivity) across the brain age models accurately predicted the age of an individual with a mean absolute error between 6.14 and 10.23 years. Brain morphometry and white matter microstructure models had a MAE of 6.76 and 7.28 respectively, which correspond with previous publications (*Cole et al., 2016*; *Han et al., 2014*; *Valizadeh et al., 2017*). In general, combining features and modalities increased the performance, and the highest performing model included a combination of both brain morphometry and white matter microstructure (mean absolute error of 6.14 years). Moreover, the correlations between the different brain age gaps suggested a relatively low level of shared variance (mean correlation = 0.51, SD = 0.13). Together these findings support the notion that tissue specific brain age models capture biologically distinct information. This is in line with the characteristic lifespan patterns of global linear decreases in gray matter volume and the nonlinear trajectories of total white matter volume and DTI based metrics of white matter microstructure (*Cox et al., 2016*; *Fjell et al., 2013*; *Ge et al., 2002*; *Liu et al., 2017*; *Raz et al., 2010*; *Westlye et al., 2010b*), highlighting that the different compartments carry unique biological information and that combining different modalities lead to a better estimation of the age of individuals (*Cherubini et al., 2016*; *Liem et al., 2017*; *Madan & Kensinger, 2018*).

## Cognitive associations

We performed a comprehensive cognitive assessment of the test sample, confirming previous evidence of substantial age-related differences in cognitive performance across
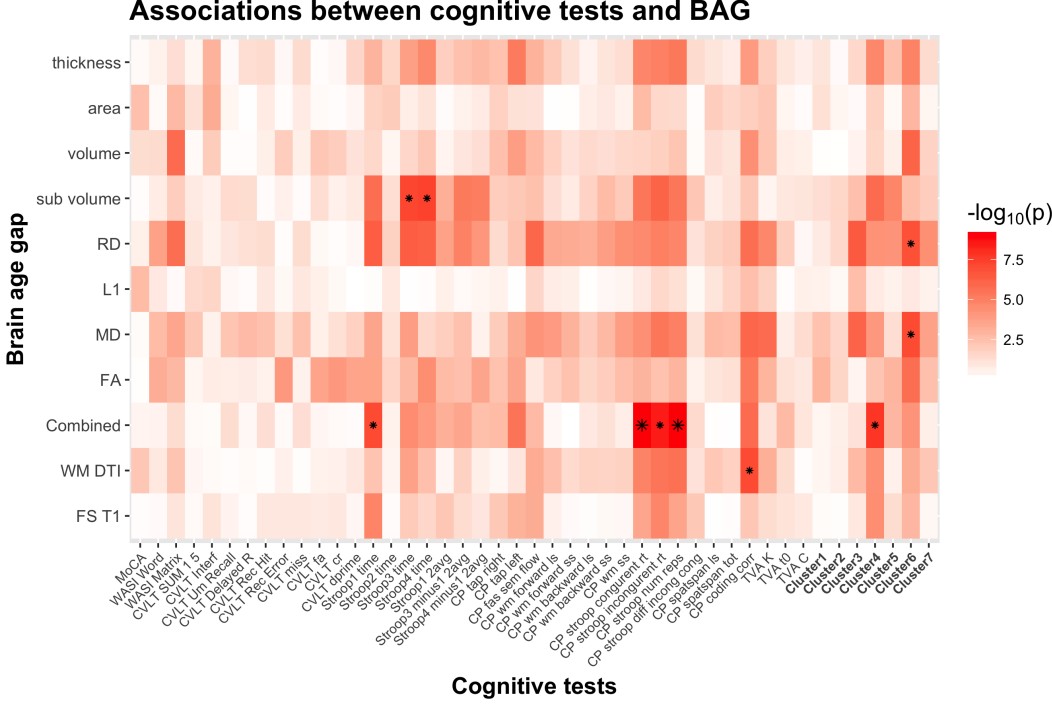

**Figure 5 Heatmap of the association between cognitive scores and brain age gaps.** The color scale depicts the minus log of the $p$-values ($-\log_{10}(p)$) for each association. The association marked with a small star represents significant associations after FDR correction, and the one marked with a big star shows significant associations after Bonferroni correction. Table S2 provides detailed overview of all abbreviations used.

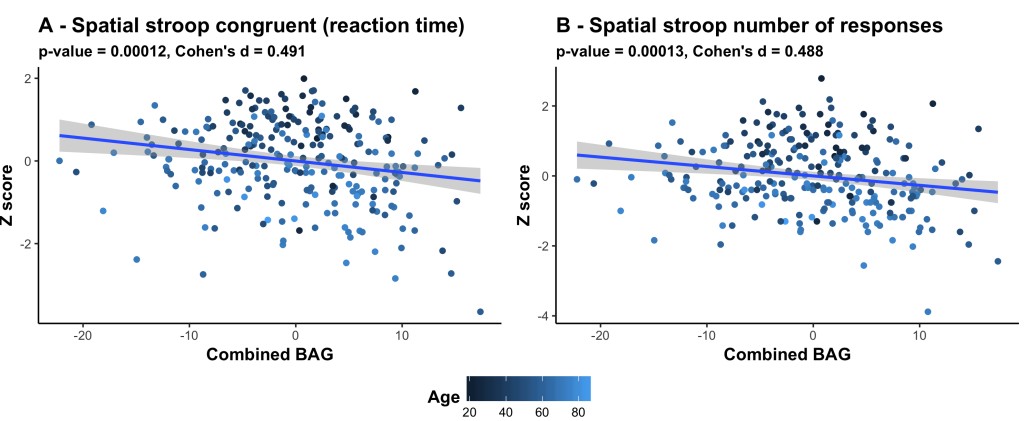

**Figure 6 Scatter plots of the 2 strongest associations between cognitive measures and BAG.** (A) Association between Spatial stroop congruent reaction time and BAG. (B) Association between Spatial stroop number of responses and BAG. The color gradient represents the age where lighter color is assigned to older individuals, and darker color to younger individuals. All associations indicate worse performance with higher brain age gap.

a range of tests and domains. Hierarchical clustering of the cognitive features indicated a characteristic pattern of covariance, largely reflecting broad cognitive domains, including memory and learning, visual processing speed, verbal skills, attentional and executive control, reasoning and psychomotor speed, and working memory. Ninety percent of the included cognitive features showed age-differences, with the largest effect sizes observed for speed-based measures, such as symbol coding test, which measures mental and visuo-motor speed (*Willer et al., 2016*). This is in line with the well-established literature on age-related decline in information processing speed in healthy aging (*Bennett et al., 2010*; *Craik & Salthouse, 2008*; *Harada, Natelson Love & Triebel, 2013*). Importantly, not only tasks measuring reaction time, but also various TVA measures based on computational modeling, such as short-term memory storage ($K$), processing speed ($C$), and perceptual threshold ($t_0$) showed strong associations with age, in line with previous studies (*Espeseth et al., 2014*; *Habekost, 2015*; *Habekost et al., 2013*; *McAvinue et al., 2012*; *Wiegand et al., 2018*).

Based on the assumption that brain age captures variance related to the integrity of the brain, we anticipated that adults with an over-estimated age would show lower cognitive performance, and that the tissue-specific brain age models would show partly differential sensitivity. To test these hypotheses, we used linear models to explore the associations between cognitive performance and BAG, with age and sex as covariates, and directly compared the parameter estimates from the different brain age models. We found a significant association between performance on several tests and BAG beyond the age associations, indicating lower performance in individuals with higher BAG. Briefly, one significant association was found for WM DTI, five for combined BAG, two for the sub-volume, one for the RD and one for the MD BAG. The strongest associations were found with the spatial Stroop congruent trials, and number of responses. These findings support that the deviance between the estimated age and the chronological age captures relevant biological information regarding the cognitive performance of an individual. Whereas we found no significantly different associations between brain age models, the association with symbol digit coding test was only seen for WM DTI BAG, while associations with Stroop 3 and 4 were observed only for sub-volume BAG, suggesting some specificity that should be investigated in future studies including larger samples and a broader spectrum of mental health, cognitive and brain phenotypes, both across healthy and clinical samples. We speculate that the contributions of the different modalities in predicting age and the associations with both cognitive performance, but also age-related illnesses vary across the age-span, as it does during maturational age (*Brown et al., 2012*). Thus, future studies might benefit from investigating modality specific brain-age estimation using specific age range, including children and adolescents.

## Limitations

The present findings do not come without limitations. First, although reducing the dimensionality of complex brain imaging data to a biologically informative brain age is a powerful method to assess deviations from normal lifespan trajectories in brain health, findings from this data reduction method are limited in specificity. Here, we attempted to both reduce the complexity of the information while keeping some modality specificity

measured by different MRI parameters. Finding a balance between specificity and precision represents an interesting challenge for future studies. Moreover, causality and individual level trajectories cannot be established based on cross-sectional data. Therefore, future longitudinal studies are needed to inform us about the relevance of the differential trajectories of the tissue-specific brain age prediction with implications for the study of a range of brain disorders. Next, although the age distribution of the test sample is irrelevant for the individual prediction accuracy, the relative overrepresentation of older individuals in the test sample is a limitation when investigating interactions between BAG and age. Thus, although the lack of brain by BAG interactions on cognitive function did not support our hypothesis that increased BAG is more relevant for cognitive function among the elderly, future studies including individuals across a broader age range and range of function are needed to characterize the lifespan dynamics in the associations between brain and behavior. More specifically, including children and adolescents would be necessary to characterize the transition between development and aging, i.e., the point of inflection from which the sign of the effects are assumed to change, an important phase that requires further investigations. Moreover, although we covered a relatively broad spectrum of structural brain features, the link between imaging based indices of brain structure and brain function is elusive, and brain age models including other brain imaging features, including functional measures, might provide a sensitive supplement to the current models. Lastly, whereas the results showed some numerical differences in the cognitive sensitivity of the different combinations of FreeSurfer based morphometry and white matter microstructure models, these differences were not statistically significant, and the hypothesis that tissue specific models provide increased specificity in terms of cognitive associations remains to be further explored in future studies.

In conclusion, we have demonstrated that models based on different combinations of brain morphometry and white matter microstructural indices provide partly differential information about the aging brain, emphasizing the relevance of tissue-specific brain age models in the study of brain and mental function in health and disease.

### Funding

This study was supported by the Norwegian ExtraFoundation for Health and Rehabilitation (2015/FO5146), the Research Council of Norway (249795, 248238), the South-Eastern Norway Regional Health Authority (2014097, 2015044, 2015073), Sunnaas Rehabilitation Hospital, and the Department of Psychology, University of Oslo. The funders had no role in study design, data collection and analysis, decision to publish, or preparation of the manuscript.

### Grant Disclosures

The following grant information was disclosed by the authors:
Norwegian ExtraFoundation for Health and Rehabilitation: 2015/FO5146.
Research Council of Norway: 249795, 248238.

South-Eastern Norway Regional Health Authority: 2014097, 2015044, 2015073.
Sunnaas Rehabilitation Hospital.
Department of Psychology, University of Oslo.

## Competing Interests

The authors declare there are no competing interests.

## Author Contributions

- Geneviève Richard conceived and designed the experiments, performed the experiments, analyzed the data, contributed reagents/materials/analysis tools, prepared figures and/or tables, authored or reviewed drafts of the paper, approved the final draft.
- Knut Kolskår, Anne-Marthe Sanders, Jennifer Monereo Sánchez, Kristine M. Ulrichsen and Erlend S. Dørum performed the experiments, authored or reviewed drafts of the paper, approved the final draft.
- Tobias Kaufmann and Nhat Trung Doan analyzed the data, contributed reagents/-materials/analysis tools, authored or reviewed drafts of the paper, approved the final draft.
- Anders Petersen analyzed the data, authored or reviewed drafts of the paper, approved the final draft.
- Dag Alnæs conceived and designed the experiments, contributed reagents/materials/-analysis tools, authored or reviewed drafts of the paper, approved the final draft.
- Ole A. Andreassen and Jan Egil Nordvik conceived and designed the experiments, authored or reviewed drafts of the paper, approved the final draft.
- Lars T. Westlye conceived and designed the experiments, analyzed the data, contributed reagents/materials/analysis tools, authored or reviewed drafts of the paper, approved the final draft.

## Human Ethics

The following information was supplied relating to ethical approvals (i.e., approving body and any reference numbers):

The study was approved by the Regional Committee for Medical and Health Research Ethics (South-East), REK 2014/694.

## Data Availability

Richard G, Westlye LT. (2018). Tissue-specific brain age prediction. Available at https://osf.io/gwqmr/.

## Supplemental Information

Supplemental information for this article can be found online at http://dx.doi.org/10.7717/peerj.5908#supplemental-information.

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
