# Peer review of "Assessing distinct patterns of cognitive aging using tissue-specific brain age prediction based on diffusion tensor imaging and brain morphometry"

_PeerJ, doi:10.7717/peerj.5908_

## Round 0.1 · original submission · Major Revisions

Dear Authors, There are major issues and numerous information/data that need to be “clear” in the revised manuscript. In particular, the comments from peer reviewer 2 and 3 need to be responded to seriously.

Reviewer 1 ·

Basic reporting

No comment

Experimental design

No comment

Validity of the findings

No comment

Additional comments

Some very minor points and recommendations
Abstract
A number of studies documented remarkably accurate age prediction, but the differential age trajectories and the cognitive sensitivity of distinct brain tissue classes ‘have to a lesser extent been
characterized’
Suggest change phrase to ‘...have yet to be adequately characterized’ to highlight the importance of this study

Introduction
Line 71 ..that individuals with an over-estimated brain age would show lower cognitive performance
Would this be true for the younger cohort?

Methods
MRI Acquisition
Line 144 following parameters:

To standardize the reporting of methods for both datasets and use the specified abbreviations after the first time, eg FOV for field of view

Discussion

350 combining different modalities lead to a better estimation the age of individuals

better estimation ‘of’ the age

Reviewer 2 ·

Basic reporting

The submitted paper is generally well conducted and asks an interesting question. More details are needed for some aspects though, such as how data was deemed to be of poor quality. Moreover, since CamCAN is a publicly available dataset, it would appreciated to appreciated if the authors could share which specific participant IDs were included/excluded--this is critical in being able to conduct follow-up research.

Experimental design

I was impressed with the experimental design and approach. This paper provides a good model of the advantages of using multiple datasets, and open data more generally. The researchers may want to comment on how their work aligns with the recommendations made in Madan (2017, Frontiers in Human Neuroscience), such as data harmonization and available metadata (e.g., since the same cognitive assessments were not available from the CamCAN dataset).

Validity of the findings

The findings appear to be well supported.

Additional comments

I was a bit disappointed to see that MRI data for the StrokeMRI study was not made available.

·

Basic reporting

The manuscript is well-written, has a good English style and uses appropriate citations. The external data sources have been sufficiently described. At the end of the introduction section (line 60) the main hypothesis is stated.

Experimental design

The experimental design is following state-of-the-art guidelines including repeated cross-validation as well as a second validation sample. However, in my opinion, the various methodological procedures and deep complexety is sometimes not described in sufficient detail to allow the general readership to gain a full picture. I would recommend to add a flow chart figure to illustrate the processes. For instance, to me it is unclear which methods have been applied before the main analysis (removing the sample differences), and which methods have been applied to obtain the final correlation/(beta?) values;and which sample has been used for the presented numbers. Additionally, the concept of generalized additive models should be briefly described to give the reader an intuition how they compare/extend linear regression models.

Validity of the findings

The authors present sound findings in predicting age from neuroimaging data, as previously described in other studies. A second aspect covered is the contribution of neurodegeneration/brain aging to predict the cognitive performance/scores. Interestingly, this is the first study to my knowledge that combines DTI and Freesurfer-derived cortical thickness for these purposes.

Additional comments

As stated above, the manuscript is well-written. However, it sometimes suffers from the large amount of information that is being presented, such that it appears a bit unfocused to me.

To refocus it a bit, I would suggest to decribe the clustering between cognitive scores (lines 282 ff) already in the methods section and to move Figure 3 to the supplementary material, as the covariance between cognitive scores does not relate to the main results presented.

Figure 1 is a bit ambigious as it is a bit unclear to me if the CAMTAN numbers are only the dark ones or if their counts start from 0. I would suggest to use two separate bars for both samples. Also, the histogram might be regenerated using 5-year intervals of age.

Fig. 2c is a bit unclear to me, as it is not described, how the composite score/correlation was derived. Further, I don't think the "on average best general predictor" is of interest as the individual scores are very heterogeneous.

Please also provide the explanation of abbreviations in the figure/table legends.

Table 1: please also report if both samples differ significantly with respect to age and gender distributions (e.g. as footnote).

---

## Round 0.2 · accepted · Accept

Dear Authors, Thank you for the revisions made that have lead to acceptance of this manuscript for publication in PeerJ.

# ·

Basic reporting

OK

Experimental design

OK

Validity of the findings

OK

Additional comments

The authors substantially revised the manuscript and adequately addressed all issues identified by the other reviewers and myself. The manuscript substantially improved soundness, comprehensibility, and readability, and from my perspective could now be accepted for publication.